# Weed Management Strategies for Tomato Plasticulture Production in Florida

**DOI:** 10.3390/plants11233292

**Published:** 2022-11-29

**Authors:** Ruby Tiwari, Mahesh Bashyal, Ramdas Kanissery

**Affiliations:** Southwest Florida Research & Education Center, Horticultural Sciences Department, University of Florida—IFAS, 2685 State Road 29 N, Immokalee, FL 34142, USA

**Keywords:** herbicides, nutsedge, plastic mulch, raised beds, row-middles

## Abstract

Florida is the top producer of fresh market tomatoes in the U.S., with an average production of 0.4 million metric tons. Tomatoes are commercially grown on plastic mulched raised beds in Southwest Florida, the primary production region in the state. Low tomato yield in plasticulture production is often associated with the poor control of nutsedge species. Nutsedge management, therefore, remains a critical production challenge for tomato growers in Florida. Sandy soil in this region promotes herbicide movement after heavy rainfall or irrigation, affecting weed suppression. This will also potentially impact the timely establishment of new tomato transplants and, consequently, the crop vigor if the herbicides get into the root zone. This review aims to present and discuss an overview of available options to safely manage major weeds of tomatoes, including nutsedge species, in plasticulture production. In addition, this review seeks to discuss an approach for utilizing herbicide adjuvants, such as spray deposition agents or oil binding agents, to improve herbicides’ efficacy and tomato crop safety by enhancing their retention in plastic mulched raised beds.

## 1. Introduction

Tomato (*Solanum lycopersicum* L.) remains the second largest agricultural crop after Citrus (*Citrus spp*.) in Florida and contributes USD 426 million to the state economy [1]. Florida’s tomato production in 2021 totaled 693 million pounds with a production value of USD 324 million. Tomatoes were harvested from 21,000 acres in 2021, representing 8% of the total U.S. acreage [2]. On average, 96% of the entire Florida tomato production was sold to the fresh market in 2020 [1]. Fresh tomatoes are produced in Florida from October to June, with peak production occurring from November to January and April to May [3].

Weed management remains a significant production challenge for tomato growers in Florida and is very important for profitable production [4,5]. Weed infestation can reduce crop yield and quality as well as increase production and harvesting costs [6]. Weed populations reduce crop yields by competing for water, light, and nutrient resources and can act as a host for nematodes, disease, and insect pests [7]. For example, broadleaf weeds, including dogfennel and Bidens (*Bidens alba*), serve as a host for sting nematodes in Florida production regions [8]. Similarly, solanaceous weeds or nightshades serve as reservoir host plants for invasive tomato leaf miner *Tuta absoluta* Meyrick (Lepidoptera: Gelechiidae) [9].

Several prior reports highlight nutsedge weed species, both yellow nutsedge (*Cyperus esculentus* L.) and purple nutsedge (*Cyperus rotundus* L.), as the most significant and widespread weed in raised bed tomato plasticulture production [10,11]. These are herbaceous perennial weeds that spread rapidly in warm regions of the world. Yellow nutsedge can populate more extensive growing areas, including temperate regions, as it can tolerate colder temperatures than purple nutsedge [12]. Both of these nutsedges species are one of the most troublesome weeds in the southern U.S. Yield losses due to competition from nutsedge has been noted in several vegetable crops, which are provided in Table 1. Similarly, the impact of nutsedge on tomato and other crops belonging to the *Solanaceae* family has been pointed out in prior studies (Table 2).

A study by Holm, Plucknett, Pancho and Herberger [12] identified purple nutsedge as the world’s worst weed, while yellow nutsedge was ranked sixteenth. Sedges are considered to be the most troublesome weed species for Florida tomato production due to the lack of herbicide options, difficult-to-reach tubers, and their ability to penetrate plastic mulches [5,13,14,15,16]. Nutsedges are problematic because they spread quickly under the plastic mulch once the rhizome develops [17]. Previous research by Webster [18] reported the production of 3500 nutsedge shoots in a 22.1 m^−2^ patch from one purple nutsedge tuber under polyethylene mulch (PE) within 60 weeks. Plastic mulch is easily punctured by nutsedges’ pointed shoot tips and therefore does not adequately control these weeds [19]. Studies by Gilreath and Santos [20], Morales-Payan, Santos, Stall and Bewick [10], and Motis, et al. [21] reported that nutsedge density, resource competition, and timing of weed emergence could negatively affect tomato production.

The effects of the nutsedge population on tomato crop yield vary with the location and production method [10,11]. A purple nutsedge density of 200 shoots m^−2^ caused a 44% yield reduction in tomatoes under greenhouse conditions [10]. By contrast, some field studies in Florida reported that nutsedge growing on the plastic mulch bed did not impact tomato yield [11,22]. However, sedges remain a significant problem in tomato plasticulture production as the wild population of nutsedges spreads rapidly, reduces crop quality, affects mulch integrity, and increases labor costs associated with harvest and mulch removal [23]. Moreover, the allelopathic potential of nutsedge tubers has been documented to adversely affect the root and shoot growth of cucumbers and tomatoes [24,25]. The association between nutsedge and soil-borne pests, such as nematodes, has also been previously described in the literature [12,26]. Schroeder, et al. [27] reported that the density of the final nematode population and yellow nutsedge tuber number showed a significant positive linear relationship, suggesting an increase in tuber number with an increased nematode population. 

**Table 1 plants-11-03292-t001:** Yield losses in vegetable crops from nutsedge interference.

Weed ^a^	Crop ^b^	Yield Loss ^c^	Reference
Purple nutsedge	Cucumber (*Cucumis sativus* L.)	43%	[28]
Purple nutsedge	Carrot (*Daucus carota* L.)	50%	[28]
Purple nutsedge	Radish (*Raphanus sativus* L.)	100%	[29]
Purple nutsedge	Lettuce (*Lactuca sativa* L.)	54%	[30]
Purple nutsedge	Garlic (*Allium sativum* L.)	52%	[28]
Purple nutsedge	Onion (*Allium cepa* L.)	89%	[28]
Yellow nutsedge	Asparagus (*Asparagus officinalis* L.)	16%	[31]
Purple nutsedge	Cilantro (*Coriandrum annum* L.)	60%	[32]

^a^ Nutsedge species. ^b^ Affected vegetable crop. ^c^ Loss in the crop yield expressed in percentage (%).

**Table 2 plants-11-03292-t002:** Yield losses in solanaceous vegetable crops due to nutsedge interference.

Weed ^a^	Crop ^b^	Yield Loss ^c^	Reference(s)
Purple nutsedge	Tomato (*Solanum lycopersicum*)	44%	[10]
Purple nutsedge	Bell pepper (*Capsicum annum* L.)	73%	[21,33]
Purple nutsedge	Tomato (*Solanum lycopersicum*)	53%	[28]
Purple nutsedge	Eggplant (*Solanum melongena* L.)	22%	[10]
Yellow nutsedge	Tomato (*Solanum lycopersicum*)	34%	[34]
Purple nutsedge *(Above-ground interface)*	Tomato (*Solanum lycopersicum*)	9%	[34]
Purple nutsedge *(Below-ground interface)*	Tomato (*Solanum lycopersicum*)	18%	[34]

^a^ Nutsedge species. ^b^ Affected solanaceous vegetable crop. ^c^ Loss in the crop yield expressed in percentage (%).

## 2. Mulching Raised Beds

Mulching in agricultural production systems has long been studied and recognized for its positive influence on the growth and yield of both annual and perennial crops [35,36]. Previous studies have reported increased tomato yield under different mulches compared to a non-mulched raised bed where weeds were not controlled (Table 3). In vegetable production, polyethylene (PE) mulches combined with pre-emergence herbicides are still the most preferred weed management method, both in the field and under a controlled environment, due to its additional benefits such as water conservation and reduction of soil compaction [37]. A study by Anzalone, et al. [38] compared nutsedge control across various mulches and reported the highest percentage of nutsedge control in PE mulch (Table 4).

Mulch can also affect insect behaviors; previous research has reported that aphid numbers were reduced by clear, black, and white P.E. mulches compared to bare soil, which was partially attributed to the suppression of weed populations [39]. In addition, reflective mulch, such as metalized plastic mulch, has been used to repel aphids, thrips, and whiteflies from colonizing young crop plants [40]. Since aphids are the primary vectors for many pathogenic viruses in crops, the tendency to attract or repel certain insects from mulch materials can be crucial for disease prevention [39,41]. 

Alternatives to using conventional non-degradable P.E. mulch include biodegradable and photodegradable options, such as plant residues and films made of paper or biodegradable polymer. These biodegradable options could significantly reduce or even eliminate costs associated with the removal and disposal of materials after crop harvest seasons [42]. Paper-based biodegradable mulches demonstrated an almost 90% reduction in weed density and biomass while also providing reasonable control of nutsedge species [43]. However, paper mulch is not as popular as P.E. mulches, as the latter offers better handling properties, durability, and ability to prevent water loss. Numerous research works have been conducted to evaluate the viability of the process of coating paper mulch with polyhydroxyalkanoates. However, this makes paper mulch heavier than its plastic counterparts, thus needing extra care and time during installation to avoid damaging the mulch [44]. Plant residues such as barley straw are also options in organic vegetable farms. Although it provides water retention and weed suppression benefits, this low-cost mulch type is not as efficient as P.E. or paper counterparts and often leads to lower yield [38,44].

**Table 3 plants-11-03292-t003:** Impact of different mulching materials on tomato yield during the fall–winter season in Brazil [45].

Mulch Types ^a^	Tomato Yield Increase ^b^	Reference
Black plastic	65.0%	[45]
White plastic	52.6%	[45]
Rice straw	47.1%	[45]

^a^ Mulch types used in tomato production. ^b^ Increase in the tomato crop yield compared to a non-mulched control, expressed in percentages (%).

**Table 4 plants-11-03292-t004:** Effectiveness of different types of mulches in controlling the purple nutsedge in tomato production systems (source: [38]).

Mulch Types ^a^	Nutsedge Control ^b^	Reference
Paper mulch	1%	[38]
Biodegradable plastic	34%	[38]
Rice straw	52%	[38]
Polyethylene mulch	81%	[38]
Maize harvest residue	52%	[38]
Barley straw	59%	[38]

^a^ Types of mulch used in tomato production for controlling nutsedge. ^b^ Nutsedge control compared to a non-mulched control, expressed in percentages (%).

The costs associated with the purchase, installation, removal, and disposal of mulch films from vegetable fields remain a major challenge when selecting mulching materials. Anderson, et al. [46] estimated that the removal cost of P.E. mulch within the United States could reach as high as USD 320/ha in the year 2021. At present, there are also growing concerns about the negative environmental impact of P.E. mulches. These P.E. mulches are made from non-renewable petroleum-based polymers. Due to strict regulations, the disposal of these mulches could pose a significant problem, as observed in European countries [43]. While the disposal cost is reduced by using biodegradable polymers, this mulching material is usually expensive. In addition, the speed of degradation is less than desirable when multiple crops are rotated seasonally on the same site [47]. Therefore, it is still more economically viable for most farms to use conventional P.E. mulch due to its lower market price. 

## 3. Controlling Weeds under Plastic Mulch

There is a diverse community of weed species in Florida’s tomato production. However, purple nutsedge and yellow nutsedge remain the most problematic weeds for tomato plasticulture production throughout Southern Florida [12]. This necessitates a well-established multi-component weed management program to ensure economical production in the region. Proper herbicide selection is key to successful weed control, and this should be based on weed type, life cycle, herbicide mode of action, and its selectivity.

In the past, tomato growers relied heavily on methyl bromide for yellow nutsedge control until its ban in 2005 by the Montreal Protocol and U.S. Clean Air Act [48]. This ban has created a void for a soil fumigant with broad-spectrum activities and consistent nutsedge control, significantly limiting cost-effective weed control options for polyethylene-mulched tomato fields [49,50]. A combination of pre-plant soil fumigants and herbicides is essential to enhance nutsedge control in polyethylene-mulched vegetable crops [16,49,51,52]. Eure and Culpepper [53] suggested that pre-plant soil fumigation followed by (fb) pre- and post-emergence herbicide applications are required to provide effective yellow nutsedge control throughout the tomato growing season. 

Tomato growers depend on fumigants, pre-emergence herbicides such as metribuzin (Mertribuzin 75 D.F. Herbicide, Makhtesim Agan of North America, Inc., Raleigh, NC, USA) and s-metolachlor, and post-emergence herbicides such as halosulfuron to control weeds. It is reported that a combination of 1,3-dichloropropene (Pic-Clor 60 Fumigant, TriEst Ag Group Inc., Greenville, NC, USA) and chloropicrin (Pic-Clor 60) (Telone C-35 Fumigant, TriEst Ag Group Inc.) is the most widely used fumigant in Florida. It can suppress weeds such as purple nutsedge [20]. Still, suppression tends to be inconsistent over time and space. Dimethyl disulfide (Paladin^®^, Arkema Inc., Exton, PA 19406, USA) provides more consistent nutsedge control than many other fumigants [54], but does not adequately control broadleaf weeds or grasses. It has also been reported that fumigation with dimethyl disulfide plus chloropicrin combined with pre-emergence s-metolachlor before polyethylene mulch installation controlled purple nutsedge; however, the control was poor during the late season [16]. 

Several pre-emergence herbicides have been registered for purple and yellow nutsedge control in tomato plasticulture production [55]. A few examples of these registered pre-emergent herbicides include fomesafen (Reflex^®^, Syngenta Crop Protection, Greensboro), sulfentrazone (Select^®^, FMC Corp., Philadelphia, PA, USA), and s-metolachlor (Dual Magnum^®^, Syngenta Crop Protection, LLC, Greensboro, NC, USA) [56,57]. These herbicides are often used in plasticulture production systems to achieve season-long nutsedge control. However, neither sulfentrazone nor s-metolachlor persists long enough in the soil to provide control of late-emerging weeds such as nutsedge [56]. *S*-metolachlor and sulfentrazone application under polyethylene mulch can alleviate the need for in-season weed control, though this observed suppression is often temporary and inconsistent across locations [11,22]. Laboratory assays by Shaner [56] reported that the half-life of sulfentrazone ranged from 121 to 302 days, while the half-life in field conditions was about two years. Moreover, Shaner [58] reported that the half-life of sulfentrazone varies from 541 days in aerobic soil to 3000 days in anaerobic soil. Similarly, Sanyal and Kulshrestha [59] reported that the half-life of s-metolachlor widely ranged from 2.5 to 289 days.

Halosulfuron (Sandea^®^, Yuman, AZ, USA) has also been recently registered for pre-transplant (sprayed on raised beds before mulch application) and post-transplant (over the top of foliage) application for nutsedge control from the four-leaf stage until about 30 days before harvesting tomatoes [56,60]. It is the most common post-emergence herbicide applied for nutsedge control in tomatoes in Florida [61]. Field observations suggest that halosulfuron application can burn down purple nutsedge foliage while effectively reducing the quantity of tuber production and viability [62]. Again, this herbicide can provide in-season weed control when applied under the polyethylene mulch, though this suppression is usually temporary and inconsistent across sites [11,22]. Besides halosulfuron, other post-directed herbicides such as imazosulfuron (League^®^, Valent U.S.A. Corporation, P.O. Box 8025, Walnut Creek, CA, USA) and trifloxysulfuron (Monument^®^, Syngenta Crop Protection, Greensboro, NC, USA) are also safe for nutsedge control on tomatoes [52]. 

Adcock, et al. [63] reported that s-metolachlor followed by (fb) halosulfuron provided reasonable control for yellow nutsedge. Similarly, a study by Devkota, Norsworthy and Rainey [57] reported a 90% reduction in yellow nutsedge with pre-emergence s-metolachlor fb post-emergence trifloxysulfuron plus halosulfuron. 

## 4. Managing Weeds between Raised Beds

In tomato plasticulture production, weeds, including broadleaf weeds and grasses, are problematic in the planting beds and on the bare ground between the plastic mulched raised beds [64]. In addition to competing with the main crop and causing yield reduction, the weeds in row middles can harbor nematodes (*Meloidogyne* spp.) and pathogens (*Phytophthora capsici*) [20,65,66,67]. 

There are some challenges associated with the pre-and post-transplanting herbicide application in vegetable row middles, specifically limited availability of registered products and the risk associated with the herbicide drift. In addition, there are only a few studies on the efficacy of herbicide application in row middles [64]. 

In a Florida study, Gilreath, et al. [68] reported 90–100% control for crabgrass [*Digitaria ciliaris* (Retz.) Koel.], pigweed (*Amaranthus viridis* L.), sida (*Sida rhombifolia* L.), and eclipta [*Eclipta alba* (L.) Hassk.] in row middles with a mix of metribuzin (Sencor^®^ DF, Bayer CropScience, Research Triangle Park, NC, USA) + cinmethylin + paraquat and oxyfluorfen + cinmethylin + paraquat compared to the weedy check. Similarly, metribuzin combined with fluazifop-p (Fusilade^®^ II, Syngenta Crop Protection, Greensboro, NC, USA) provided 70–100% weed control compared to the weedy check. However, cinmethylin is not registered for use in tomatoes in Florida [64]. 

Recent research in Florida showed no difference in weed density reduction when a post-emergence herbicide carfentrazone (Aim^®^ 2EC, FMC Corp., Philadelphia, PA, USA) was applied without pre-emergence herbicides compared to the non-treated control plots. In contrast, paraquat, another post-emergence herbicide, reduced total weed density by 67% in the spring and 61% in the fall seasons when applied without pre-emergence herbicides [64]. Similarly, when the carfentrazone was tank mixed with pre-emergence herbicides such as flumioxazin or s-metolachlor, and when paraquat was tank mixed with flumioxazin, s-metolachlor, or metribuzin, there were 81 to 90% fewer broadleaf weeds, including wild radish (*Raphanus raphanistrum* L.), Florida pusley [*Richardia scabra* (Moq.) Gomez], cutleaf primrose (*Oenothera laciniata* Hill), and lambsquarters (*Chenopodium album* L.), compared to the untreated control. Furthermore, pre-emergence s-metolachlor alone or tank mixes containing s-metolachlor reduced grass density by 62 to 88% compared to untreated control during spring in vegetable row middles [64]. Previous studies have reported a similar level of grass control with s-metolachlor alone or combined with flumioxazin [69,70]. 

A listing of various herbicides used for the weed control in Florida tomato plasticulture production is provided in Table 5.

## 5. Environmental Fate of Herbicides Utilized in Tomato Production

The fate and persistence of herbicides in the soil are determined by several factors, such as microbial degradation, chemical decomposition, adsorption on soil colloids, leaching, volatility, photodecomposition, and uptake by plants [72] (Figure 1). 

Halosulfuron falls in the category of herbicides (mode of action group 2) that interfere with the enzyme acetolactate synthase (ALS). Acetolactate synthase synthesizes branched-chain amino acids, specifically valine, leucine, and isoleucine, which are essential for protein synthesis and plant growth [73]. These herbicides are absorbed in plant roots and foliage and readily translocated in the xylem and phloem. Sulfentrazone belongs to a group of herbicides known as Protoporphyrinogen Oxidase (PPO) inhibitors (mode of action group 14). These herbicides inhibit the activity of the enzyme protoporphyrinogen oxidase, which plays a crucial role in chlorophyll biosynthesis [73]. Moreover, these herbicides cause a significant increase in the level of porphyrin, which leads to rupture of plant cell membranes.

Microbial degradation is a major metabolic pathway of sulfonylurea herbicides degradation in the soil [74]. Microorganisms, primarily algae, fungi, actinomycetes, and bacteria, feed on organic matter, including organic herbicides, for energy and growth [72]. Increased persistence of sulfonylurea herbicides such as halosulfuron have been reported in soil of pH 7.0 and higher, with half-lives extending into years [75]. This is primarily attributed to an increase in anionic forms of sulfonylureas under basic soil conditions, resulting in decreased microbial degradation and dissipation. Soil adsorption for sulfonylurea herbicides such as halosulfuron has been reported to be negatively correlated with increasing pH and positively correlated with increased organic matter [75]. Similarly, Grey, et al. [76] indicated that halosulfuron dissipation was more rapid for bare soil than under low-density polyethylene mulch, which can probably be attributed to reduced microbial activity inside polyethylene mulches.

S-metolachlor is a very-long-chain fatty acid (VLCFA)-inhibiting herbicide. This group of herbicides interferes with the enzyme VLCFA synthase, which catalyzes the formation of long-chain fatty acids [77]. They impair cell division and shoot development in susceptible plants [78]. Bedmar, et al. [79] reported DT_50_ (time to reach 50% of initial concentration) for s-metolachlor to range from 82 to 141 days for A to C soil horizons for Typic Agriudolls soils in Argentina. Moreover, a negative correlation between DT_50_ values of s-metolachlor with soil organic carbon content has been reported in the literature [79,80]. Corroborating these findings, a recent study by Marín-Benito, et al. [81] reported slower dissipation and leaching of s-metolachlor in soils amended with green compost and pelletized manure compared to unamended soils. DT_50_ values of metolachlor were prolonged from 37.9 to 126 days to 49.5 to 135.9 days for two different soil types under soil sterilization in China, indicating that microbial degradation was the dominant pathway for metolachlor and s-metolachlor degradation in soils [80,82]. Similarly, several other studies [83,84] highlighted the role of microorganisms in mineralizing metolachlor by using herbicide as a sole source of carbon and energy for growth.

Sulfentrazone mobility in soil is affected by soil texture, with lesser mobility in clay loam (R_f_ or herbicide retention factor, ranging from 0.3 to 0.4) and greater mobility in loamy sand (R_f_ ranging from 0.5 to 0.8) [85]. Additionally, sulfentrazone movement was reported to be greater in soils with high pH and coarse texture such that the order of adsorption to soil was Sequatchie loam > Dothan loamy sand > Malden loamy sand > Commerce silty clay loam > Harkey clay loam [86]. Previous studies have reported DT_50_ for sulfentrazone to vary from 24 to 113 days in fine loamy soil in Tennessee and a half-life of 146.5 days on Typic Hapludox soils in Brazil [87,88]. Microbial degradation is considered the primary method of sulfentrazone dissipation in the soil, as reviewed by Martinez, Silva, Fay, Maia, Abakerli and Durrant [87]. It is likely that soil moisture and temperature also impact the degradation rate of sulfentrazone through the effect of these parameters on soil microbial activity. The majority of soil microorganisms are dormant at 4.4 °C, while temperatures ranging from 24 °C to 32 °C favor microbial activity [72]. Moreover, microbial activity is greater in warm, moist, well-aerated, and fertile soil and can quickly decompose these organic herbicides.

## 6. Improving Herbicide Use in Tomato Plasticulture Production

One of the major challenges associated with the chemical weed control options is a steady loss of active herbicide ingredients (a.i.) from the weed seed germination zone in soil [89,90,91]. Moreover, most pre-emergence herbicides do not control nutsedge species in tomato plasticulture production due to herbicide ineffectiveness and crop phytotoxicity [11,92]. Therefore, alternative and crop-safe weed management strategies are crucial to protect tomato crops from herbicide injury and provide adequate nutsedge suppression in tomato plasticulture production.

Adjuvants are soil and spray deposition agents for improving the efficacy of soil-applied herbicides. They also improve soil adsorption by keeping the a.i. in the weed suppression zone for longer, minimizing the risk of leaching into the tomato root zone where there is potential for crop injury (Figure 2) [93]. In some herbicides, adjuvants are already included in the formulations for sale, such as glyphosate (Roundup PowerMax^®^, Bayer Crop Sciences, Saint Louis, MO, USA), or they may be purchased separately and mixed with herbicides before use [94]. There are two main types of adjuvants categorized by their function, i.e., activator adjuvants and utility adjuvants. Activator adjuvants enhance an herbicide’s performance by increasing their absorption rate into targeted crops [95]. They increase the herbicide formulation’s ability to kill targeted weed species without affecting crops [96,97,98]. Utility adjuvants, also known as spray modifiers, alter herbicide dynamics (physical and chemical characteristics of spray mix) to improve application efficiency and provide adequate weed control [95]. They also alter the herbicide’s formulation to cover plant surfaces evenly, keeping it in contact with plant tissues without runoff [99]. Adding an appropriate adjuvant can reduce the herbicide application and minimize the costs for weed control [100].

A previous greenhouse study at the University of Florida (UF) by Abouziena, et al. [101] evaluated varying rates (0.84, 1.12, 1.68, and 2.24 kg ha^−1^) of bentazon (Basagran, BASF Corporation, 26 Davis Drive, Research Triangle, NC 27709, USA) effects alone or in combination (tank mixed) with the adjuvants ammonium sulfate (AMS), Induce (Induce, Helena Chemical Company, 225 Schilling Blvd., Collierville, TN 38017, USA), and Kinetic (Kinetic, Helena Chemical Company, Memphis, TN 38137, USA) on common cocklebur (*Xanthium strumarium* L.), black nightshade (*Solanum nigrum* L.), velvetleaf (*Abutilon theophrasti* Medik), and strangler vine [*Morrenia odorata* (Hook. & Arn.) Lindl.]. Black nightshade, a hard-to-control weed, was poorly controlled (<55%) at all rates of herbicide treatments, with or without adjuvants [101,102]. However, bentazon at its lowest rate plus adjuvant easily suppressed common cocklebur [101]. Similarly, a study conducted in far northern Queensland (Australia) reported that adding the oil-based adjuvant Grounded^®^ (Helena Agri Enterprises, LLC) to a spray tank mixture of pre-emergence herbicides did not provide effective weed control in ratoon cane [103]. However, soil samples taken from the topsoil in the same study indicated a higher percentage of herbicide after rainfall events in the plots receiving the tank mix (herbicide plus Grounded^®^) compared to the control (without adjuvant) [103]. A preliminary study conducted at UF Southwest Florida Research and Education Center, Immokalee, FL, during the spring and fall of 2021, reported consistent reduction in purple nutsedge density when the pre-emergence herbicide s-metolachlor was mixed with Grounded^®^ and sprayed on tomato beds under plastic mulch (unpublished data).

## 7. Conclusions and Future Directions

Yellow and purple nutsedge has been a major challenge for successful tomato plasticulture production in Florida. Their rhizomes produce multiple shoots with pointed tips that pierce plastic mulch, making harvest difficult. In addition, these weeds compete with the tomato crop for light and nutrient sources and can harbor nematodes or pathogens. Chemical weed control is a vital tool to control nutsedge spp. under plastic mulch. Because of cost-effectiveness, tomato growers primarily depend on herbicides for weed management. Many pre-emergence herbicides have been registered for nutsedge control under plastic mulch, yet their effectiveness has been inconsistent over time and space. Therefore, a combination of pre- and post-emergence herbicides have been mainly recommended for controlling nutsedge spp. 

Pre-emergence herbicide application under plastic mulch may have a wide range of unintended consequences on tomatoes. Sandy soil of Florida, with its low organic content, increases risk of herbicide leaching into the crop’s root zone, potentially diminishing vigor of tomato crop transplants under plastic mulch. Applying a pre-emergence herbicide in combination with a soil-binding agent or adjuvant on raised bed plastic mulch could be a crop protection measure, while also limiting herbicide movement and prolonging the persistence of herbicide a.i. in soil, thereby providing adequate nutsedge suppression.

In summary, the best strategy to achieve season-long nutsedge control combines pre-emergence herbicides with well-planned post-transplant herbicide application [63,104]. Integrating pre-emergence herbicides with post-emergence applications would significantly reduce nutsedge density and biomass production, resulting in higher tomato yields. Moreover, combining herbicides with a soil-binding agent could be more effective than using chemicals only for Florida sandy soils. Future research to identify the most effective post-transplant herbicides for controlling nutsedge in plastic mulch tomato production would provide more tools to growers. With herbicide leaching posing a significant problem in Florida’s sandy soil, more studies are needed to understand the mobility and persistence of pre- and post-emergence herbicides under tomato plastic mulch beds.

## Figures and Tables

**Figure 1 plants-11-03292-f001:**
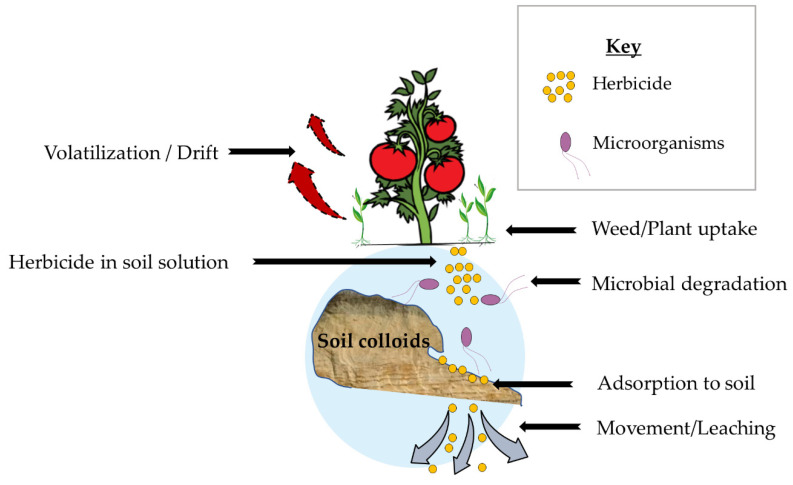
Schematic representation of factors determining herbicide fate in soil.

**Figure 2 plants-11-03292-f002:**
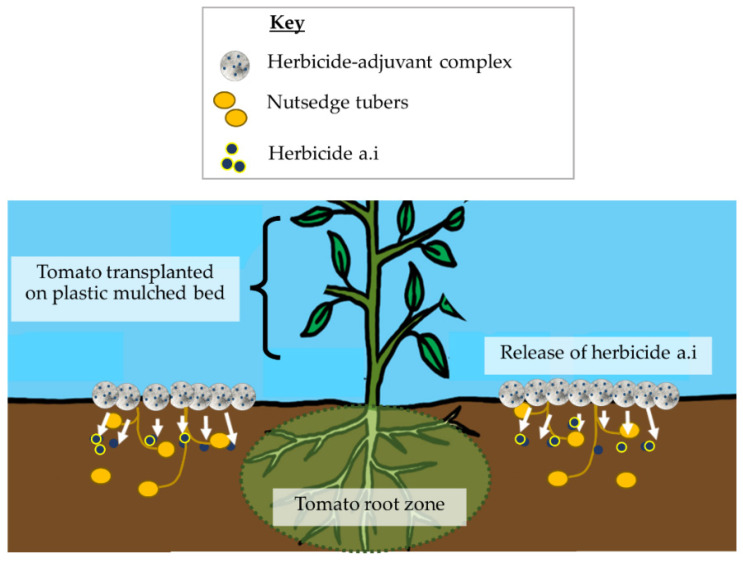
Schematic illustration of adjuvant use with soil-applied herbicides for effective weed suppression and enhanced crop safety.

**Table 5 plants-11-03292-t005:** Chemical weed control options in tomato plasticulture production systems. Source: [71].

Herbicide a.i. ^a^	Trade Name ^b^	Mode of Action Group	Application Timing ^c,d^	Rate (lbs. a.i./acre)
Carfentrazone	(Aim^®^) 2 EC or 1.9 EW Up to 2 fl. oz.	14	Post-emergence	0.031
Clethodim	SelectMax	1	Post-emergence	0.9–2.5
Glyphosate	Roundup	9	Pre-transplant; Post-emergence	0.3–1
EPTC	Eptam^®^ 7E	8	Pre-transplant	2.62–3.5
Halosulfuron	(Sandea^®^, Profine^TM^) 75 DG	2	Pre-transplant; Post-emergence	0.024–0.036
Lactofen	Cobra^®^	14	Post-emergence	0.25–0.5
S-metolachlor	Dual Magnum^®^	15	Pre-emergence	1–1.3
Metribuzin	Sencor DF	5	Pre-transplant	0.25–0.5
Paraquat	Gramoxone	22	Post-emergence	0.62–0.94
Trifluralin	Treflan HFP	3	Pre-transplant	0.5
Oxyfluorfen	(Goal^®^) 2 XL 1–2 pt. (GoalTender^®^) 4 E 0.5–1 pt.	14	Pre-transplant	0.25–0.5
Flumioxazin	(Chateau^®^) 51 WDG Up to 4 oz.	14	Pre-emergence	Up to 0.128
Pendimethalin	(Prowl^®^ H_2_O) 3.8 1.0–1.5 pt.	3	Pre-transplant	0.48–0.72
Sulfentrazone	Spartan FL 4F 2.25–6.0 fl. oz.	14	Pre-transplant	0.07–0.19
Diquat	(Reglone^®^) 1.5 pt.	22	Post-emergence	0.38
Imazosulfuron	(LeagueTM) 0.5 DF 4–6.4 oz.	2	Post-emergence	0.19–0.3
Rimsulfuron	(Matrix^®^ FNV, Matrix^®^ SG, PruvinTM) 25 WDG 1.0–2.0 oz.	2	Pre-emergence or post-emergence	0.02–0.03
Sethoxydim	(Poast^®^) 1.5 EC 1.0–1.5 pt.	1	Post-emergence	0.19–0.28
Trifloxysulfuron	(Envoke^®^) 75 DG 0.1–0.2 oz.	2	Post-transplant	0.0047–0.0094
Fomesafen	(Reflex^®^) 2 SC 1–1.5 pt.	14	Pre-transplant	0.25–0.38
Napropamide	(Devrinol)50 DF 2–4 lb.	15	Pre-transplant	1–2

^a^ a.i.: active ingredient. ^b^ Including but not limited to these products. ^c^ Pre- and post-transplant indicates before and after tomato transplant, respectively. ^d^ Pre- and post-emergence indicates before and after weed emergence, respectively.

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
