# Peer review of "Weed Management Strategies for Tomato Plasticulture Production in Florida"

_plants, 2022, doi:10.3390/plants11233292_

Round 1

Reviewer 1 Report

Dear Editor,

the review manuscript " Weed Management Strategies for Tomato Plasticulture Production in Florida" by Tuwari and colleagues present and discuss an overview of available options to manage major weeds of tomatoes in plasticulture production.

In its present form, the manuscript is well readable and the informations presented by the authors are easy to follow.

Manuscript could be accepted in the present form.

Author Response

Thank you for this comment. We appreciate you for taking the time to review our manuscript.

Reviewer 2 Report

I carefully analyzed the review paper. I believe that the topic is particularly important and has the potential to contribute to the development of knowledge.

However, I consider that the paper is poorly documented and, above all, it does not sufficiently synthesize the specialized literature to present a proper review.

I recommend that the important themes, such as production, the degree of weed control, on different types of soil, etc. in relation to the types of mulch to be presented as bibliographic summaries presented in tables.

I also recommend that each chapter, even if it does not have tables, should present results, ideas synthesized from the bibliographic analysis carried out. By this I mean that a string of aspects is not enough, but you also need your own contribution in the paper.

Reviewer 3 Report

The subject of article is interesting. In many regions of world where tomatos are produce this weed control could be problematic. For this reason we need to search for rational solution and also economic reasonable methods for obtain higher productivity. The manuscript is written in a clear way, easy to read and with appropriate reference to data in table and figures.

 In my opinion presented solutions should be more precise. The entire article including the summary and the title should be rewritten regarding the described weed species particularly threatening the tomato crop, since in the vast majority of the Authors refer to the impact of one species. Please revise also the literature list as required by the journal.

All suggestions are included in the manuscript.

Reviewer 4 Report

The paper is interesting and well written. Such review papers are needed in all the fields for summarizing the available information from a scientific field at a proper moment in time.

Author Response

(The authors gave the same response as above.)

Round 2

Reviewer 2 Report

The manuscript can be published in present form.